# Efficient Domain Continual Pre-training by Mitigating the Stability Gap

## Abstract

Adapting Large Language Models (LLMs) to specialized domains like medicine and law through domain continual pre-training has become the cutting-edge method. However, contrary to our expectations of immediate gains, we've uncovered a surprising phenomenon: a temporary performance drop at the start of the process, followed by a performance recovery phrase. This drop is not only unexpected but remarkably consistent across different model sizes and domains, such as medical and law. To gain a deeper understanding of this issue, we introduce the concept of stability gap—borrowed from visual models dealing with new class classifications—to explain this initial drop in LLM performance. Based on this concept, we hypothesize that the initial performance drop arises from instability in the model's general abilities, which we further validated through our experiments. We further reveal that this initial instability is intricately tied to training settings that involve distribution shifts. To address this initial instability and enhance LLM performance within a fixed compute budget, we propose one training strategy that reduces the instability by increasing the epoch number, along with two data sampling strategies focused on data quality and corpus distribution. We conduct various experiments on Llama-family models to validate the effectiveness of our strategies in both medical and legal continual pre-training and instruction tuning. For example, our strategies improve the average medical task performance of the OpenLlama-3B model from 36.2% to 40.7% with only 40% of the original training budget and enhance the average general task performance without causing forgetting. Furthermore, we apply our strategies to continually pre-train and instruction-tune the Llama-3-8B model. The resulting model, **Llama-3-Physician**, achieves the best medical performance among current open-source models and performs comparably to or even better than GPT-4 on several medical benchmarks.

## 1 Introduction

Continual pre-training is an important approach for LLMs to improve their performance in target domains (Huang et al., 2023; Yang et al., 2024a; Chen et al., 2023c), learn new topics and languages (Jiang et al., 2024; Gupta et al., 2023), and even boost their general capabilities (Ibrahim et al., 2024). While extensive research has focused on understanding LLM mechanisms during pre-training from scratch (Biderman et al., 2023a; Xue et al., 2024), far less attention has been given to how LLMs behave during continual pre-training (Que et al., 2024). This gap in the literature is particularly striking given the importance of continual pre-training in adapting models to new domains and evolving knowledge. In this paper, we report a surprising phenomenon observed during continual pre-training: rather than an immediate improvement, LLM performance on target domain tasks initially declines in the early stages of training. Only after further training, when more data is incorporated, does performance recover and eventually surpass that of the original model. We consistently observe this performance pattern—a V-shaped curve—across various model scales and target domains, including both medical and legal fields. This counterintuitive finding challenges common assumptions about continual pre-training, where improvement is typically expected at initial.

To explore the underlying mechanisms of this phenomenon, we draw inspiration from the concept of the stability gap (De Lange et al., 2022; Caccia et al., 2021), originally introduced in the context

of vision models in continual learning. The stability gap describes how a model's performance on previously learned tasks initially degrades when learning new tasks, before gradually recovering as it adapts. Previous research attributes this initial drop to an imbalance between the model's stability gradient—its ability to maintain performance on prior tasks—and its plasticity gradient—the capacity to adapt to new ones. Early in training, the model's plasticity gradient dominates, leading to a temporary performance decline. As training progresses, the stability gradient strengthens, allowing performance to recover.

Applying this framework to LLMs, we hypothesize that the initial performance drop in continual pre-training stems from a similarly insufficient stability gradient to preserve the model's general capabilities (e.g., instruction-following skills). Over time, as the plasticity gradient diminishes and the stability gradient rises, task performance rebounds. Supporting this hypothesis, we observe a similar V-shaped pattern in general-domain tasks, where initial declines give way to recovery. Further analysis of weight updates throughout the training process provides additional evidence for this interpretation.

But how can we harness it to optimize continual pre-training? Given a fixed computing budget, we know that the stability gap causes inefficiency in continual pre-training as it delays performance improvement. To address this, we propose three efficient continual pre-training strategies:

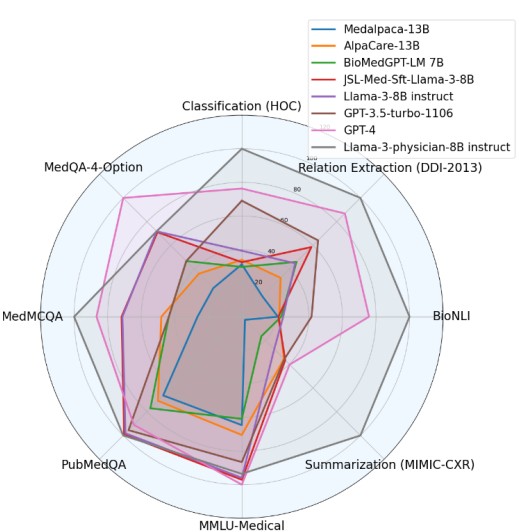

1. Instead of continually pre-training the LLM on a large corpus for one epoch, which induces a large plasticity gradient for a long period, we continually pre-train the LLM on a subset of the corpus with a proper size for multiple epochs.

2. Select the subset with the highest-quality tokens to learn rich domain knowledge, leading to faster performance recovery and higher peak performance.

3. Use a data mixture that is similar to the pre-training data distribution in data source and rate, thus reducing the distribution shift and mitigating the knowledge forgetting of general instruction-following ability.

Figure 1: The performance comparison between our model (Llama-3-physician) and other baselines involves reporting the ratio of each model's task performance to the best performance of that task among all models.

To verify our strategy, we first conduct experiments on the OpenLlama-3B model with medical and legal domain continual pretraining. We find that our strategies not only accelerate performance improvement by mitigating the stability gap but also improve the LLM's peak performance. We also compare our strategies with other continual pre-training techniques and analyze the influence of important learning factors, such as learning rate, for our strategies in Sec. 5. Finally, we apply our strategies to both the continual pretraining and instruction tuning processes of the Llama-3-8B model (Meta, 2024), efficiently enhancing its performance on diverse medical tasks, outperforming other open-source LLM baselines, and achieving performance comparable to GPT-4 (See performance comparison in Figure 1).

## 2 RELATED WORK

**Large language Models** such as GPT-4 (OpenAI, 2023), Gemini (Team), and Llama (Touvron et al., 2023a)), have billions of parameters and show strong performance on various basic natural language tasks (Qin et al., 2023), human examination (Hendrycks et al., 2020b; Zhong et al., 2023), and agent-related tasks (Guo et al., 2023; Liu et al., 2023; Zhou et al., 2023). Their success attracts

researchers to analyze LLMs' learning properties during the pre-training process (Kaplan et al., 2020; Biderman et al., 2023a; Zhang et al., 2024a). Kaplan et al. (2020) finds the pre-training scaling rule for model size and dataset size and then Hoffmann et al. (2022) proposes the Chinchilla rule that claims the equal importance of the model size and the number of training tokens. Sorscher et al. (2022) further claims that pruning low-quality data can improve the above neural scaling laws. However, high-quality training tokens are limited and may be run out soon (Villalobos et al., 2022). Thus, some researchers try to maximize the utilization of the existing corpus by training it for multiple epochs (Muennighoff et al., 2024; Xue et al., 2024). But they observe the performance degradation (Hernandez et al., 2022; Xue et al., 2023; Hoffmann et al., 2022) after training 4 epochs.

**Continual pre-training** gradually becomes necessary for LLMs to expand their basic ability (Wu et al., 2022; Fu et al., 2024; Zhuang et al., 2024), avoid outdated information (Jiang et al., 2024), and become the domain expert (Huang et al., 2023; Yang et al., 2024a; Chen et al., 2023c; Nguyen et al., 2023; Wu et al., 2023; Yıldız et al., 2024; Xie et al., 2024a). The domain corpus for continual pre-training can be collected by n-gram models (Muennighoff et al., 2024), heuristic rules designed by human experts (Chen et al., 2023c; Zhang et al., 2024c) or automatically identified by a LLM (Zhang et al., 2024c). For the continual pre-training techniques. Ke et al. (2023; 2022) focused on adding masks or adjusting the architecture of small Language models like RoBERT to protect the learned general knowledge. However, these techniques result in huge computational consumption for LLMs. Recent studies (Gupta et al., 2023) show that learning rate re-warming can improve LLMs' downstream task performance and a stability gap appears when replaying the previous data. Ibrahim et al. (2024) further claims that learning rate re-warming, re-decaying, and replay can make the continual pre-training performance match the performance of fully re-training when continually pre-training the English LLM on the German corpus. Other continual pre-training method studies focus on selecting useful tokens (Lin et al., 2024), expanding MOE architecture (Chen et al., 2023a), and knowledge distillation (Jin et al., 2021b).

**Continual learning and the Stability Gap** Continual learning aims to design methods that can learn new knowledge without the catastrophic forgetting of previously learned knowledge (Kirkpatrick et al., 2017; Van de Ven et al., 2022). To mitigate the forgetting problem when learning a new task, replaying previous tasks' data (Rolnick et al., 2019; Buzzega et al., 2020; Prabhu et al., 2020; Buzzega et al., 2021; Guo et al., 2022) becomes the main approach. De Lange et al. (2022); Caccia et al. (2021) further find that, although they conduct the replay approach, the vision model still first loses its performance stability in previous classification tasks ( the performance drops abruptly) and then gradually recovers. They call it the stability gap phenomenon. Different from them, we focus on the continual pre-training of the LLM and observe that both the LLM's domain task performance and general ability suffer from the stability gap.

## 3 IDENTIFYING THE STABILITY GAP IN CONTINUAL PRE-TRAINING

In this section, we describe the unique performance phenomenon observed during continual pre-training, where performance on the target domain initially drops before rising. We then introduce the concept of the stability gap to explain this behavior and validate our explanation through experiments.

### 3.1 INVESTIGATING THE BEHAVIOR OF LLMS DURING CONTINUAL PRE-TRAINING

**Experiment setup** In this study, we chose OpenLlama3B-v2 (Geng & Liu, 2023) as our default LLM and use the medical domain as our primary target domain. Following previous work (Chen et al., 2023b), we set the compute budget to 50 billion (50B) training tokens. To collect the continual pre-training corpus, we follow the simple and scalable methodology of Muennighoff et al. (2024); Lin et al. (2024). First, we train a small model (e.g., KenLM (Heafield, 2011)) on a high-quality medical reference corpus. Then, we use the trained small model to calculate the perplexity (PPL) of samples in the Refined-Web dataset (Penedo et al., 2023). Finally, we extract 50B tokens from the Refined-Web dataset with the lowest PPL to create the medical corpus. More details are provided in Appendix A.

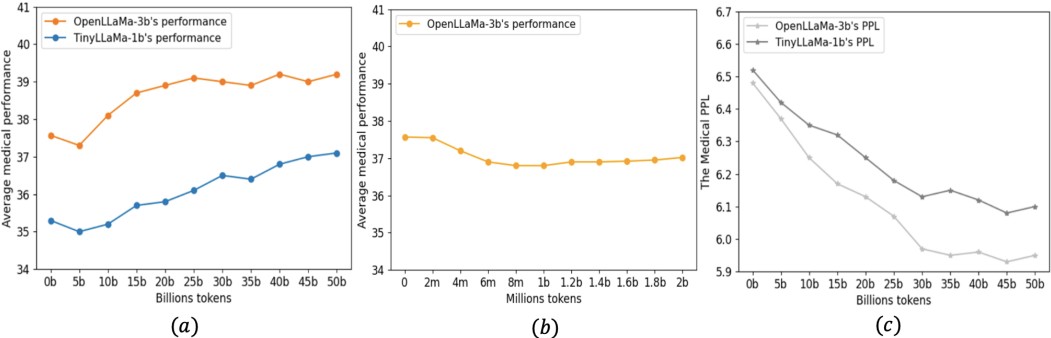

Figure 2: (a) reports the models' average medical performance during the medical continual pre-training process. (b) reports the models' average medical performance at the beginning. (c) illustrates the models' average medical perplexity (PPL) during the medical continual pre-training process.

**Observation (1): The medical task performance first drops and rises during continual pre-training.** Specifically, we follow Chen et al. (2023c) and measure the average accuracy performance over the MMLU-Medical-Genetics (Hendrycks et al., 2020a), MedQA (Jin et al., 2021a), PubMedQA (Jin et al., 2019), and MedMCQA (Pal et al., 2022) tasks (see task details in Appendix C). We report the average performance on medical tasks every 5 billion training tokens. From Figure 2(a), we observe that the domain task performance initially drops during the first 5 billion tokens and then gradually recovers and improves. Furthermore, as shown in Figure 2(b) (a fine-grained view), we observe that the performance declines sharply at the beginning, followed by a gradual recovery. Additionally, we consider the TinyLlama model (Zhang et al., 2024b), a 1.1B Llama model trained on 3 trillion tokens, and continually pre-train it on the medical corpus. From Figure 2(a), we observe that its performance on medical tasks also shows the same trend, despite being trained on so many tokens.

**Observation (2): The perplexity of medical Wikipedia steadily declines during continual pre-training.** We further measure the average perplexity (PPL) of the models on the Wikipedia corpus about medical terms[1]. From Figure 2(b), we observe that the PPL steadily drops. This indicates that the LLM has acquired medical domain knowledge at the initial continual pre-training and continues improving its medical domain knowledge throughout the entire continual pre-training process.

**More Observations:** We also examine continual pretraining in both the legal domain and a general setting. Similar V-shaped performance curves are observed, reinforcing that the initial performance drop followed by a subsequent rise in target task performance is a common phenomenon in the continual pretraining of LLMs. Detailed results are provided in Appendix B.

## 3.2 STABILITY GAP: A CONCEPTUAL EXPLANATION FOR THE INITIAL PERFORMANCE DROP AND THEN FOLLOWING RECOVERY.

**The Stability Gap** refers to the initial decline in a vision model's performance on previous tasks while learning a new task, followed by a subsequent improvement, even when data from the earlier tasks is replayed. Lange et al. (2022) explains this by disentangling the model gradient $\mathcal{G}$ into $\alpha$-weighted plasticity and stability components: $\mathcal{G} = \alpha\mathcal{G}_{plasticity} + (1 - \alpha)\mathcal{G}_{stability}$, where $\mathcal{G}_{plasticity}$ focuses on learning the new task by minimizing its data loss, while $\mathcal{G}_{stability}$ seeks to maintain performance on previous tasks by keeping the loss of replay data low. They attribute the initial performance drop to the plasticity gradient exceeding the stability gradient to reduce new task loss, resulting in a failure to maintain performance on previous tasks. As performance declines, the stability gradient strengthens, leading to a balance between gradients and eventual performance recovery.

**Explanation of our observations** Directly applying the concept of the stability gap to explain our phenomenon is not feasible, as we do not replay the pretraining corpus. However, during domain-

---

[1]https://huggingface.co/datasets/gamino/wiki_medical_terms

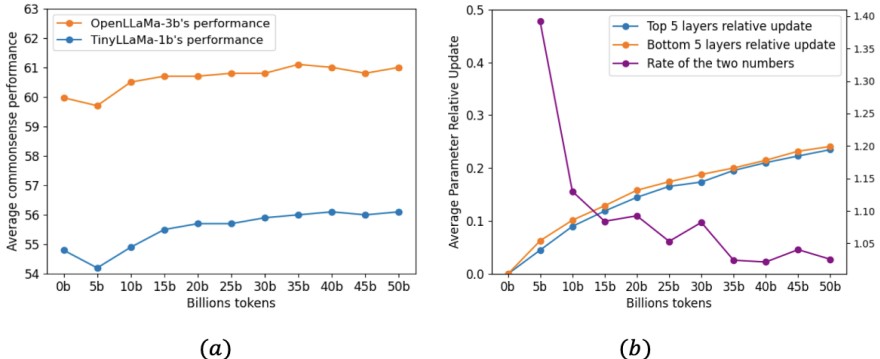

$(a)$ $(b)$

Figure 3: (a) shows the OpenLLaMa's average common-sense task performance during medical continual pre-training. (b) illustrates the OpenLlama model's relative parameter update during the medical continual pre-training process. We report the average weight relative update of weights in the top 5 layers and the bottom 5 layers. We also report the rate between the two average numbers.

specific continual pretraining, the language modeling loss serves two critical functions: it explicitly learns domain-specific knowledge while implicitly preserving general knowledge and text modeling capabilities, as the domain corpus still contains general information. This implicit preservation acts as a form of 'self-replay', providing the stability gradient. Further, we infer that performance declines because the plasticity gradient for learning domain-specific knowledge surpasses the stability gradient for retaining general text knowledge and text modeling ability. Over time, the stability gradient strengthens to restore general knowledge and text modeling abilities, while the plasticity gradient has learned knowledge in the target domain, leading to performance improvement.

**Empirical verification for our explanation**    Based on our inference, we can predict that the general task performance follows a similar V-shape curve as the stability gradient gradually rises. We verify our prediction in Figure 3(a). We also find evidence for our explanation at the weight level by (2) measuring the relative weight update of each weight $w$ as $\frac{w_t - w_0}{w_0}$, where $w_t$ is the weight value during continual pre-training and $w_0$ is the original weight value. A high relative weight update indicates a large gradient for updating the weight. Figure 3(b) shows that the bottom layers' weights initially have a higher relative weight update than the top layers ($rate > 1.35$). Previous studies indicate that bottom layers learn the syntax and low-level semantics (Devlin et al., 2019; Hewitt & Manning, 2019; Ling et al., 2023), while top layers contain high-level semantics and task-specific knowledge (Yang et al., 2024b; Chen et al., 2024). This suggests that the top layers' weights indeed lack sufficient stability gradient to maintain instruction-following ability initially. The performance then recovers as the relative weight updates (stability gradient) increase in the top layers and domain knowledge is learned, as indicated by the continuous drop in medical perplexity.

## 4    EFFICIENT CONTINUAL PRETRAINING STRATEGIES FOR MITIGATING THE STABILITY GAP

In this section, we propose three efficient continual pre-training strategies for reducing the above stability gap problem. The training process and details follow those in the above section.

**Strategy I: Continually pre-train the LLM on a corpus subset across multiple epochs rather than the entire large corpus for a single epoch.**    The key insight is that a larger corpus demands a high plasticity gradient for a longer period. In contrast, pretraining the LLM on a properly sized random subset of the corpus across multiple epochs reduces the need for sustained high plasticity after the first epoch and accelerates the rise of the stability gradient. In Figure 4(a), we observe that this strategy leads to faster performance recovery. The LLM achieves peak performance at the fourth epoch, consistent with previous studies (Xue et al., 2024).

**Strategy II: Continually pre-train the LLM on the corpus subset with the highest quality.**    The performance of domain tasks also depends on the learned domain knowledge. Therefore, collecting

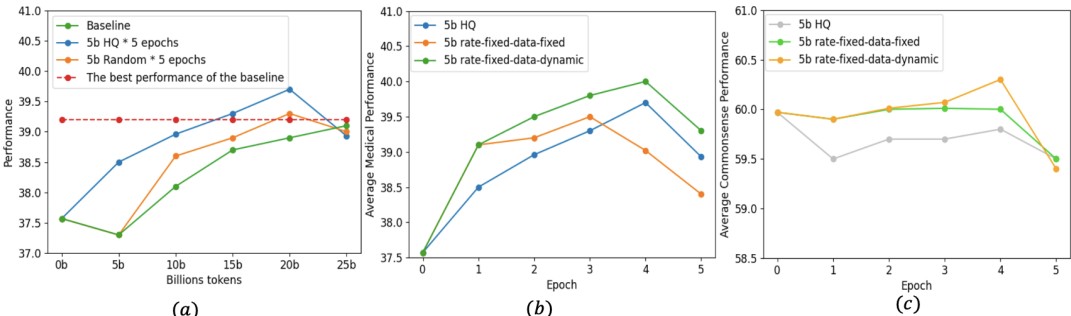

(a)  (b)  (c)

Figure 4: (a) reports the average medical performance during the medical continual pre-training process. The baseline is pre-training the OpenLlama-3B model with 50b medical tokens with one epoch. '5b Random' is pre-training the LLM with 5b tokens randomly selected from the 50b medical tokens for 5 epochs. '5b HQ' is pre-training the LLM with the highest quality (HQ) 5b tokens of the 50b medical tokens for 5 epochs. (b) shows the average medical performance across 5 epochs. (c) illustrates the average commonsense task performance across 5 epochs.

a subset with the highest quality should further enhance performance. To verify this, we used the trained KenLM from Sec. 3.1 to calculate the perplexity (ppl) of each sample in the entire medical corpus. A lower perplexity indicates that the sample is closer to the distribution of the medical reference corpus. We then continually pre-trained the OpenLlama-3B model on the subset with the lowest perplexity (i.e., the highest quality) for multiple epochs. From Figure 4 (a), we observe that the high-quality subset indeed enables the LLM to recover performance faster and stronger in the medical domain. Further analysis of the pre-training subset size is presented in Sec. 5.2.

**Strategy III: Use a data mixture rate similar to the pre-training data.** The pre-training data mixture rate is a vital factor for the pre-training performance of large language models (LLMs) (Xie et al., 2024c; Shen et al., 2023). Therefore, we propose a third strategy that follows the pre-training data's mixture rate to construct the continual pretraining training subset, aiming to reduce the distribution gap and stabilize the instruction-following ability of the LLM during continual pre-training. Specifically, for the OpenLlama model, we follow the Llama mixture rate (Touvron et al., 2023a) to collect 5 billion tokens initially. We then replace the CC and C4 data (82% of the 5 billion tokens) with medical tokens sampled from the highest quality 5 billion medical tokens (HQ-5b). There are two ways to sample these medical tokens. The first method randomly samples the medical tokens once to construct a fixed training corpus. We call this "rate-fixed-data-fixed". The second method randomly samples the medical tokens from the HQ-5b tokens for each epoch. We call this "rate-fixed-data-dynamic".

From Figure 4(b), we observe that the second method achieves a higher peak performance as it offers a better trade-off between recovering performance and learning domain knowledge. Additionally, our strategies further improve the average performance on general commonsense tasks, as shown in Figure 4(c), and reduce the medical perplexity and the rate of relative weight update, as detailed in Appendix D. We also investigate the effectiveness of our three strategies in the general continual pre-training setting in Appendix E.

## 5 EVALUATION

In this section, we first compare the effectiveness of our strategies with other continual pre-training techniques. Next, we investigate the impact of important learning factors, such as the learning rate, on our strategies. Finally, we deploy our strategies into the newest Llama-3-8b model, which achieves the strongest fine-tuned performance among open-source baselines.

### 5.1 COMPARISON WITH OTHER CONTINUAL PRE-TRAINING TECHNIQUES

**Baselines and evaluation tasks** We consider the following baselines for comparison: (1) Continually pre-training the OpenLLaMa-3B LLM with 50 billion collected medical tokens for one epoch ("the full token baseline"). (2) Re-warming and re-decaying the learning rate of (1) based

on the paper by (Ibrahim et al., 2024). (3) Replay baselines: Following (Chen et al., 2023b), we randomly sample 5B (10%), 10B (20%), and 15B (30%) tokens from OpenLLaMa-3B's pretraining dataset (the RefinedWeb dataset) and combine them with 50B medical tokens. Pretraining is stopped once a total of 50B tokens have been processed. This baseline does not consider the data mixture rate. (4) Parameter protection baselines: Following (Harun & Kanan, 2023), we freeze the top 5 layers' weights during the continual pre-training process of (1) to protect the high-level instruction-following ability and mitigate the stability gap. We also consider another baseline that freezes the bottom 5 layers' weights for comparison. We follow (Chen et al., 2023b) and consider the tasks of PubMedQA, MedMCQA, and MedQA-4-Option. For the MMLU benchmark (Hendrycks et al., 2020a), we consider the average performance of its medical topics, including medical genetics, anatomy, clinical knowledge, professional medicine, and college medicine. We use the lm-evaluation-harness framework (Gao et al., 2023) to measure the baselines' zero-shot performance. The training details are the same in Appendix A.

| Method | Training tokens number | MMLU-Med-Avg | PubMedQA | MedMCQA | MedQA-4-Option | Avg |
|---|---|---|---|---|---|---|
| OpenLLaMa-3B | - | 25.6 | 68.4 | 25.4 | 25.4 | 36.2 |
| Full token baseline | 50B | 26.1 | 70.4 | 26.1 | 27.1 | 37.4 |
| Re-warming and re-decaying | 50B | 26.5 | 70.3 | 27.1 | 27.1 | 37.7 |
| Replay 5B data | 50B | 26.3 | 69.2 | 27.6 | 26.9 | 37.5 |
| Replay 10B data | 50B | 29.3 | 71.0 | 30.4 | 27.6 | 39.5 |
| Replay 15B data | 50B | 29.0 | 70.1 | 29.4 | 26.2 | 38.7 |
| Freezing top 5 layers | 50B | 26.2 | 69.9 | 27.1 | 27.3 | 37.6 |
| Freezing bottom 5 layers | 50B | 26.0 | 69.1 | 25.4 | 25.7 | 36.5 |
| Our strategies | 20B | **30.0** | **71.2** | **34.0** | **27.8** | **40.7** |

Table 1: Zero-shot accuracy across various medical benchmarks.

**Results** From Table 1, we find that (1) our strategies improve the base model's average medical task performance significantly (4.5%) with only 20 billion training tokens. This demonstrates the effectiveness and efficiency of our strategies for continual pre-training. (2) Other techniques can also improve continual pre-training performance, except for the baseline 'Freezing bottom 5 layers,' which hinders the learning of medical domain knowledge. We further verify our strategies' effectiveness in continual law pretraining. We put the results in Appendix F.

### 5.2 FACTOR ANALYSIS

**Impact of learning rate and training subset size** To analyze the impact of training factors such as learning rate and training subset size, we conducted a series of experiments, with details provided in Appendix G. Our findings show that a learning rate that is too high leads to significant drops in generalization ability, while a rate too low hampers the acquisition of new domain knowledge. Additionally, using a subset that is too large (e.g., 10 billion tokens) introduces a stability gap and slows performance. Conversely, a smaller subset yields better initial performance but leads to rapid overfitting in later epochs. Finally, we validate the optimal hyperparameter configuration for our experiments.

### 5.3 DEPLOYING OUR STRATEGIES INTO THE LLAMA-3 MODEL

**Continual pre-training** We continually pre-train the Llama3-8B-base model using our three strategies with the high-quality 5 billion medical tokens constructed in Sec. 4 for 4 epochs. The training details are in Appendix H. After the continual pre-training process, we find that the average medical performance drops slightly, likely due to the unknown data mixture rate of Llama-3 and the lack of access to its high-quality pre-training corpus for performance recovery. However, the medical perplexity is significantly lower than that of the Llama3-8B-base model.

**Task-specific fine-tuning** To evaluate LLMs' performance in the supervised learning setting, we follow (Chen et al., 2023b) and individually conduct task-specific finetuning on both the base models and the continually pre-trained models using each benchmark's training set. We also consider 8 task-finetuned baselines. We put task details in Appendix C and training and baseline details in Appendix H.

| Model | MMLU-Medical | PubMedQA | MedMCQA | MedQA-4-Option | Avg |
|---|---|---|---|---|---|
| Llama-2-7B (Touvron et al., 2023b) | 56.3 | 61.8 | 54.4 | 49.6 | 53.2 |
| BioMistral SLERP 7B (Labrak et al., 2024) | 60.5 | 75.2 | 44.2 | 47.3 | 56.8 |
| MEDITRON-7B (Chen et al., 2023b) | 55.6 | 74.4 | 59.2 | 52.0 | 57.5 |
| Llama3-Aloe-8B-Alpha (Gururajan et al., 2024) | 72.7 | 77.2 | 59.0 | 62.3 | 67.8 |
| Llama-2-70B | 74.7 | 78.0 | 62.7 | 61.3 | 67.2 |
| MEDITRON-70B | 73.6 | **80.0** | 65.1 | **65.4** | 69.0 |
| GPT-3.5-turbo-finetuned (Shi et al., 2024) | 70.5 | 71.4 | 61.8 | 63.3 | 66.7 |
| Llama-3-8B Fine-tuned (ours) | 82.3 | 75.8 | 60.0 | 61.1 | 69.8 |
| Llama-3-8B Full (ours) | 82.0 | 78.6 | 61.8 | 60.8 | 70.8 |
| Llama-3-Physician-8B (ours) | **85.0** | 79.1 | **81.4** | 61.5 | **76.7** |

Table 2: Accuracy comparison across various medical benchmarks in the task-specific fine-tuning setting. Llama-3-8B Fine-tuned is directly fine-tuned on these tasks. For 'Llama-3-8B Full', we first continually pre-trained the Llama with 50B medical tokens and then finetuned the pretrained model on these tasks. For Llama-3-Physician-8B, we first continually pre-trained the Llama with with our strategies and then finetuned the pretrained model on these tasks.

**Results** We use the lm-eval-harness (Gao et al., 2023) to evaluate our model (Llama-3-Physician) and related baselines' performance. No demonstration examples are used. From Table 2, we find that our model outperforms other baselines with similar model scales on the four evaluation benchmarks by a clear margin. This is due to the following reasons: (1) we use the newest and strongest open-source Llama-3 model rather than older Llama-2 or Mistral-7B, (2) we continually pre-train the base model with high-quality medical tokens (compared to 'Llama-3-8B fine-tuned and Llama-3-8B instruct'), and (3) our strategies further boost the gains from continual pre-training markedly (compared to 'Llama-3-8B Full'). Our model also outperforms many larger LLMs (70B) on average, meaning that users can obtain higher-quality medical services with a faster inference rate and less memory consumption.

## 5.4 DEPLOYING OUR STRATEGIES INTO THE INSTRUCTION TUNING PROCESS

Instruction-tuning is an important approach to boost the LLM's performance among multiple tasks. We follow Xie et al. (2024b) and consider the instruction-tuning setting that tunes the continual pretrained Llama-3-8B model (see the above section) with a combination of medical tasks. More training details are in Appendix H.

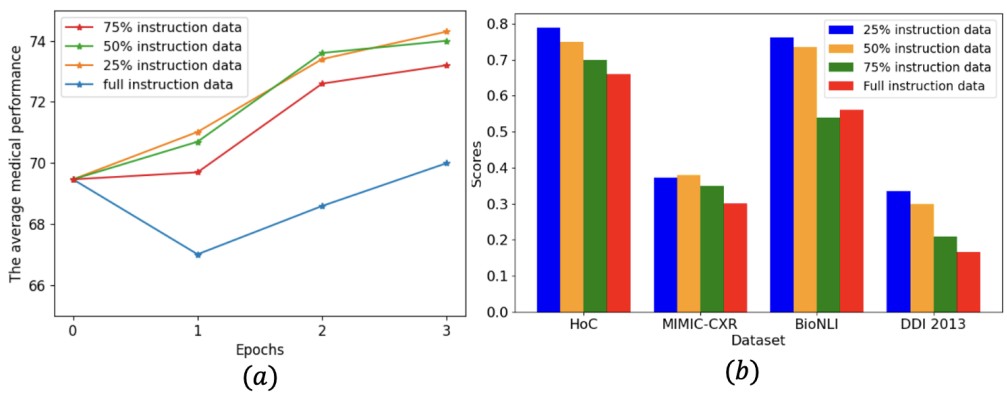

$(a)$ $(b)$

Figure 5: We consider the 'full instruction data' experiment as fine-tuning the model with all instruction data for 3 epochs. For the '$n$% data' experiments, we first uniformly sampled the highest quality instructions from each instruction dataset based on scores provided by the Deita data selector. We then mixed the sampled data with the general instructions from the Airoboros-3.2 dataset. The total training tokens are equal to $n$% of the full instruction data. We set $n$ to 25, 50, and 75 here. (a) shows the experiments' average medical question-answering task performance during instruction tuning. (b) illustrates the experiments' performance for other medical tasks. For BioNLI, DDI 2023, and HOC tasks, we report macro-F1 as the score. For MIMIC-CXR summarization tasks, we report Rouge-L as the score.

**Deployment** In the instruction tuning process, our first strategy is common as the medical instruction tuning process usually involves multi-epochs training (Zhang et al., 2023a; Xie et al., 2024b; Han et al., 2023). For the second strategy, we consider Deita (Liu et al., 2024), a simple automatic instruction data selector, to select high-quality medical instruction data. This selector uses the LLM to give quality scores for instructions and considers the diversity of instruction data by sampling data from different clustering. For the last strategy, we consider high-quality general instruction datasets like Airoboros-3.2 Durbin (2024) to mitigate the forgetting in general instruction following ability.

**Observations** From Figure 5, we first observe that the average performance of medical question-answering tasks initially drops slightly (in the first epoch) and then gradually rises, which is similar to the phenomenon observed in the continual pre-training process. Additionally, we observe that our strategies can mitigate the initial performance drop and achieve higher peak performance during the instruction tuning process, thereby extending the application of our strategies. Figure 5 also shows that we only need computation equivalent to 25% of the original instruction data (consisting of high-quality medical instruction data and general instruction data) to achieve the best performance among diverse tasks. This reduces computational consumption and improves the efficiency of the instruction tuning process. We call the tuned model in the experiment '25% instruction data' as 'Llama-3-physician-8B instruct'. In the following paragraphs, we will compare it with other baselines.

**Baselines** For instruction-tuning, we consider instruction-tuned models like Mistral-7B-instruct (Jiang et al., 2023), Zephyr-7B-$\beta$-instruct (Tunstall et al., 2023), PMC-Llama-7B (Wu et al., 2023), BioMedGPT-LM 7B (Zhang et al., 2023a), Medalpaca-13B (Han et al., 2023), AlpaCare-13B (Zhang et al., 2023b), Me-LLaMA-13B chat(Xie et al., 2024b), Llama-3-8B instruct (Meta, 2024), and JSL-Med-Sft-Llama-3-8B (johnsnowlabs, 2024). These LLMs are tuned with general instructions or medical task instructions.

| Model | MMLU-Medical | PubMedQA | MedMCQA | MedQA-4-Option | Avg |
|---|---|---|---|---|---|
| Mistral-7B-instruct (Jiang et al., 2023) | 55.8 | 17.8 | 40.2 | 41.1 | 37.5 |
| Zephyr-7B-instruct-$\beta$ (Tunstall et al., 2023) | 63.3 | 46.0 | 43.0 | 48.5 | 48.7 |
| PMC-Llama-7B (Wu et al., 2023) | 59.7 | 59.2 | 57.6 | 49.2 | 53.6 |
| Medalpaca-13B (Han et al., 2023) | 55.2 | 50.4 | 21.2 | 20.2 | 36.7 |
| AlpaCare-13B (Zhang et al., 2023b) | 60.2 | 53.8 | 38.5 | 30.4 | 45.7 |
| BioMedGPT-LM 7B (Zhang et al., 2023a) | 52.0 | 58.6 | 34.9 | 39.3 | 46.2 |
| Me-Llama-13B (Xie et al., 2024b) | - | 70.0 | 44.9 | 42.7 | - |
| Llama-3-8B instruct | 82.0 | 74.6 | 57.1 | 60.3 | 68.5 |
| JSL-Med-Sft-Llama-3-8B (johnsnowlabs, 2024) | 83.0 | 75.4 | 57.5 | 59.7 | 68.9 |
| GPT-3.5-turbo-1106 | 74.0 | 72.6 | 34.9 | 39.3 | 60.6 |
| GPT-4 (OpenAI, 2023) | **85.5** | 69.2 | 69.5 | **83.9** | **77.0** |
| Llama-3-physician-8B instruct (ours) | 80.0 | **76.0** | **80.2** | 60.3 | 74.1 |

Table 3: Accuracy comparison for question-answering tasks in the instruction-tuning setting.

**Results** From Table 3, we find that our model outperforms other open-source baselines in question-answering tasks by a clear margin. Additionally, our model's average performance is close to that of GPT-4. Furthermore, in Table 6, we observe that our model significantly outperforms GPT-4 in medical classification, relation extraction, natural language inference, and summarization tasks. This demonstrates the significant advantage of our model in processing diverse medical tasks.

## 6 CONCLUSION

Our paper explores the behavior of LLMs when continually pre-training them on a new domain's corpus and observes the stability gap, a phenomenon marked by a significant initial performance drop followed by a slow recovery. We explain it from the view of plasticity and stability gradients and then propose three strategies that effectively improve the LLM's domain performance and reduce computational costs by reducing the stability gap. Furthermore, we deploy our strategies on the newest Llama-3-8B model, which achieves the strongest performance among open-source baselines of similar model scales and outperforms the closed-source GPT-3.5 model.

**Limitations and Potential impacts** Ideally, knowing the pre-training data mixture could maximize the outcome of our method, but most strong open-source LLMs didn't provide their training data mixture. Our Llama-3-8B experiment shows we can still improve significantly in this scenario. Due to limitations in computing resources, we plan to verify our conclusions and strategies on larger LLMs in the future. Our strategies are designed to address the machine learning problem of the stability gap, and we do not see any potential risks. The datasets and base models used in this paper will be open-sourced. Although we do not consider our model to be ready for real-world medical use in its current form, we are releasing it to the research community to promote work on large language models for the medical domain and the safety of language models in medical applications.

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

## A   THE DETAILS OF PRE-TRAINING

For OpenLLaMa-3B, TinyLLaMa-1B, and Pythia-410m, we download them from their official website. For OpenLLaMa-3B and TinyLLaMa-1B LLMs, we continually pre-train them with the 50 billion medical tokens we constructed in Sec. 4 for one epoch. For the high-quality medical reference file, we use the dataset 'wiki_medical_terms' downloaded from the huggingface. For the high-quality legal reference file, we use the dataset 'Caselaw Access Project' downloaded from the huggingface. For the Pythia-410m LLM, we continually pre-train it with the 100 billion tokens randomly sampled from the 2021-2022 subset of the RefinedWeb dataset. We consider this subset as the Pile dataset only contains data before the year 2021 and then the tokens sampled from the 2021-2022 subset are unseen for the Pythia-410m model. The pre-training code is based on the transformers. The task is to predict the next token with a context size of 2048. The training is executed using 192 V100 GPUs. We employ the AdamW optimizer with $\beta_1 = 0.9, \beta_2 = 0.95$, a weight decay of 0.01, and a learning rate of 3e-4. We use a cosine learning rate scheduler with a 0.1 warmup ratio for gradual adaptation to training complexity and bf16 precision for computational efficiency. Gradient accumulation is set to 4 steps, and each training batch contains about 340 million tokens. We also add support for FlashAttention-2 (Dao, 2023) for more efficient inference and long-context decoding.

When deploying our strategies into the continual pretraining process, we use the same learning rate schedule as the one used for pretraining. For baselines, we follow their setups in their official papers.

## B   MORE OBSERVATION AND ANALYSIS

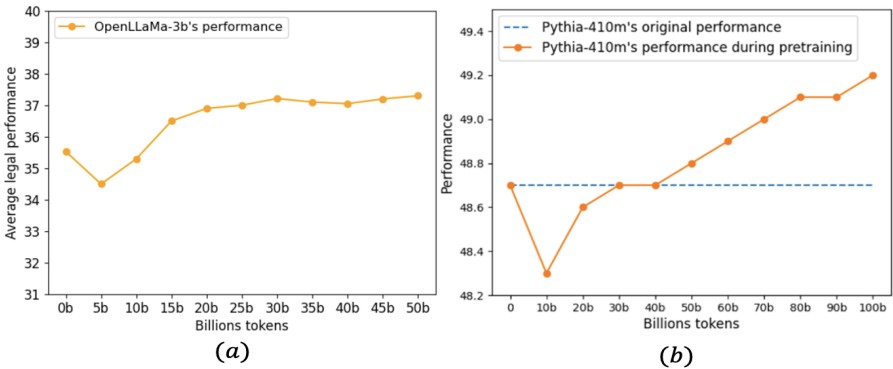

Figure 6: (a) shows the OpenLLaMa's average legal task performance during law continual pretraining. (b) illustrates the OpenLlama model's relative parameter update during the medical continual pre-training process. We report the average weight relative update of weights in the top 5 layers and the bottom 5 layers. We also report the rate between the two average numbers.

For continual law pretraining, we use the same procedure to collect domain corpus and the same optimization setup to train the LLM. For its evaluation, we consider three QA tasks: MMLU-International-Law, MMLU-Professional-Law, and Contract-QA from LegalBench Guha et al. (2023). We report the average performance in Figure 6(a), which shows a similar v-shape performance curve. Continual pretraining on another large corpus is an important approach to boost the pretrained LLM's general task performance (Jiang et al., 2024; Gupta et al., 2023). We call it the general continual pretraining setting. We further find that it also exists a similar performance phenomenon. Specifically, we continually pre-train the Pythia-410m model (Biderman et al., 2023b) (initially pre-trained on the Pile (Gao et al., 2020) dataset) on the RefinedWeb dataset (Penedo et al., 2023) to boost its general ability. We measure its general ability using the average performance across 10 common-sense tasks and report the average performance of every 10 billion tokens. Training details are in Appendix A and task details are in Appendix C. From Figure 6(b), we observe that the LLM's general task performance first drops significantly and then gradually rises.

Based on our observations, the initial drop followed by a rise in target task performance is a general phenomenon in the continual pre-training of LLMs of various sizes.

## C  TASK AND BASELINE INFORMATION

For the medical evaluation, we follow Chen et al. (2023b) and mainly consider the following four tasks:

**MedMCQA** (Pal et al., 2022) is a large-scale and comprehensive dataset for multichoice (four-option) medical question answering. It is derived from real-world medical entrance exam questions (Indian AIIMS and NEET-PG) and consists of over 194,000 high-quality medical questions. These questions cover 2,400 healthcare topics and 21 medical subjects, exhibiting a wide range of topical diversity. The average token length is 12.77.

**MedQA** (Jin et al., 2021a)is a multichoice question-answering dataset collected from the professional medical board exam, the United States Medical License Exams (USMLE). It comprises 12,723 questions sourced from a comprehensive collection of 18 English medical textbooks that have been extensively utilized by medical students and USMLE candidates. Questions in MedQA cover a wide range of topics in clinical medicine, necessitating responses with professional expertise and complex multi-hop reasoning across multiple pieces of evidence. The average question and option length is 116.6 and 3.5, respectively.

**MMLU** (Hendrycks et al., 2020b) is a comprehensive multi-task language understanding test dataset that encompasses 57 tasks across various domains such as mathematics, history, computer science, law, etc. In our experiments, we specifically focus on a subset of medical reasoning-related tasks including clinical knowledge, college medicine, medical genetics, and professional medicine.

**PubMedQA** (Jin et al., 2019) is a biomedical question and answering dataset derived from PubMed abstracts. It contains 1k expert annotated multi-choice question-and-answer samples based on 211.3k PubMed articles. The task of PubMedQA is to provide answers to research questions with yes/no/maybe responses based on the corresponding abstracts. The average question and context length is 14.4 and 238.9, respectively.

**HOC** (Baker et al., 2016) is a classification task to decide the Hallmarks of Cancer (HOC) taxonomy of the article based on its abstract. The input is an abstract text. There are 10 topics you will need to decide whether the article is related to. Topics: sustaining proliferative signaling, evading growth suppressors, resisting cell death, enabling replicative immortality, inducing angiogenesis, activating invasion and metastasis, genomic instability and mutation, tumor promoting inflammation, and cellular energetics, and avoiding immune destruction.

**DDI 2023** (Segura-Bedmar et al., 2013) is a task to predict the relationship between the given head entity labeled as $@DRUG1$ $and$ $tail$ $entity$ $labeled$ $as$ $@DRUG2$ within a given sentence, this relation which must be in ('mechanism', 'effect', 'advice', 'int', 'none'). mechanism: this type is used to annotate drug-drug interactions that are described by their pharmacokinetic mechanism. effect: this type is used to annotate drug-drug interactions describing an effect or a pharmacodynamic mechanism. advice: this type is used when a recommendation or advice regarding a drug interaction is given. int: this type is used when a drug-drug interaction appears in the text without providing any additional information. none: there are no drug-drug interactions.

**BioNLI** (Bastan et al., 2022) is a task to classify the relationship between the given medical premise and hypothesis into one of the following labels: entailment, contradiction, or neutral. This dataset contains abstracts from biomedical literature and mechanistic premises generated with nine different strategies.

**MIMIC-CXR** (Johnson et al., 2019) is a generation task that derives the impression from findings in the radiology report.

The dataset statistics are in Table 4

For the evaluation of general task ability, we consider the following 10 commonsense tasks:

**ARC-Challenge and ARC-Easy** ARC (Clark et al., 2018) is a multiple-choice question-answering dataset, containing questions from science exams from grade 3 to grade 9. The dataset is split into two partitions: Easy and Challenge, where the latter partition contains the more difficult questions that require reasoning. Most of the questions have 4 answer choices.

Table 4: Dataset statistics

| Dataset | # Train | # Test | Source |
|---|---|---|---|
| MedMCQA (Pal et al., 2022) | 182,822 | 4183 | Exam |
| MedQA (Jin et al., 2021a) | 10178 | 1273 | Exam |
| MMLU (Hendrycks et al., 2020b) | - | 163 | Exam |
| PubMedQA (Jin et al., 2019) | 211,269 | 500 | Literature |
| HOC (Baker et al., 2016) | 1108 | 315 | Literature |
| DDI 2023 (Segura-Bedmar et al., 2013) | 1108 | 315 | Literature |
| BioNLI (Bastan et al., 2022) | 5544 | 6308 | Literature |
| MIMIC-CXR (Bastan et al., 2022) | 122,014 | 1606 | Literature |

**BoolQ** (Clark et al., 2019) is a question-answering dataset for yes/no questions containing 15942 examples. These questions are naturally occurring —they are generated in unprompted and unconstrained settings. Each example is a triplet of (question, passage, answer), with the title of the page as optional additional context. The text-pair classification setup is similar to existing natural language inference tasks.

**COPA** (Roemmele et al., 2011) consists of 1000 questions, split equally into development and test sets of 500 questions each. Each question is composed of a premise and two alternatives, where the task is to select the alternative that more plausibly has a causal relation with the premise.

**HellaSWAG** (Zellers et al., 2019) is a dataset for studying grounded commonsense inference. It consists of 70k multiple choice questions about grounded situations: each question comes from one of two domains – activitynet or wikihow – with four answer choices about what might happen next in the scene. The correct answer is the (real) sentence for the next event; the three incorrect answers are adversarially generated and human-verified, so as to fool machines but not humans.

**OpenBookQA** (Mihaylov et al., 2018) is a new kind of question-answering dataset modeled after open-book exams for assessing human understanding of a subject. It consists of 5,957 multiple-choice elementary-level science questions (4,957 train, 500 dev, 500 test), which probe the understanding of a small "book" of 1,326 core science facts and the application of these facts to novel situations.

**PIQA** (Bisk et al., 2020) dataset introduces the task of physical commonsense reasoning and a corresponding benchmark dataset Physical Interaction: Question Answering or PIQA. Physical commonsense knowledge is a major challenge on the road to true AI-completeness, including robots that interact with the world and understand natural language. PIQA focuses on everyday situations with a preference for atypical solutions.

**Race** (Lai et al., 2017) is a large-scale reading comprehension dataset with more than 28,000 passages and nearly 100,000 questions. The dataset is collected from English examinations in China, which are designed for middle school and high school students. The dataset can serve as the training and test sets for machine comprehension.

**SciQ** (Welbl et al., 2017) dataset contains 13,679 crowdsourced science exam questions about Physics, Chemistry and Biology, among others. The questions are in multiple-choice format with 4 answer options each. For the majority of the questions, an additional paragraph with supporting evidence for the correct answer is provided.

**WinoGrande** (Sakaguchi et al., 2021) is a new collection of 44k problems, inspired by the Winograd Schema Challenge (Levesque, Davis, and Morgenstern 2011), but adjusted to improve the scale and robustness against the dataset-specific bias. Formulated as a fill-in-a-blank task with binary options, the goal is to choose the right option for a given sentence which requires commonsense reasoning.

We use the lm-eval-harness (Gao et al., 2023) to evaluate the LLM on these tasks' test set and report the zero-shot performance.

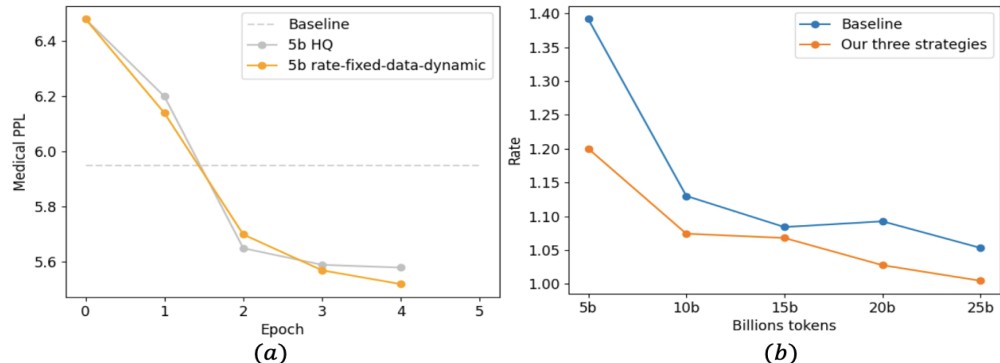

Figure 7: (a) reports the average medical perplexity of the OpenLLaMa-3B using our strategies. '5b HQ' means the LLM using our strategies I and II. '5b rate-fixed-data-dynamic' means the LLM using our three strategies. 'Baseline' is the average medical perplexity of the OpenLLaMa-3B model that has been continually pre-trained with 50 billion medical tokens. (b) shows the rate between the bottom 5 layers' average relative parameter and the top 5 layers' average relative parameter update of the OpenLLaMa-3B using our strategies. 'Baseline' is the rate of the OpenLLaMa-3B model during the continual pre-training with 50 billion medical tokens.

## D   THE PERPLEXITY AND RELATIVE PARAMETER UPDATE RATE OF THE LLM USING OUR STRATEGIES

From Figure 7(a), we observe that the LLM using our strategies gradually decreases its average medical perplexity, indicating that the LLM is acquiring rich medical knowledge. Its average medical perplexity at the fourth epoch is even lower than that of the OpenLLaMa-3B model, which has been continually pre-trained with 50 billion medical tokens. From Figure 7(b), we also find that the ratio between the average relative parameter updates of the bottom 5 layers and the top 5 layers of the OpenLLaMa-3B model using our strategies is closer to 1. This suggests that the plasticity gradient and the stability gradient are more balanced when employing our strategies.

## E   DEPLOYING OUR STRATEGIES INTO THE GENERAL CONTINUAL PRE-TRAINING SETTING

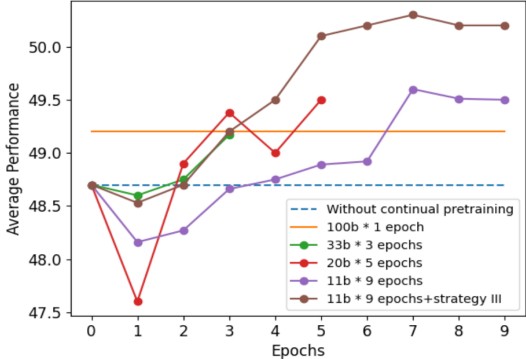

Figure 8: We report the average performance of the 10 commonsense and reading compression task here. The Model is Pythia-410m.

Continually pre-training one LLM on another large corpus is an approach to boost its general ability (Gupta et al., 2023). We consider the scenario of continually pre-training the Pythia-410m model on the RefinedWeb dataset. The Pythia-410m model has been pre-trained on the Pile dataset. In this context, we use the average performance of 10 commonsense and reading comprehension tasks, as detailed in Appendix C, to measure the LLM's general task performance. To test the effectiveness of strategy I in the general continual pre-training setting, we conduct multi-epoch experiments with different training subset sizes. The tokens in each training subset are randomly sampled from the

RefinedWeb dataset and the computational consumption of each experiment can not be beyond the compute budget (100 billion tokens). From Figure 8, we find that strategy I indeed helps the Pythia-410m model to mitigate the stability gap and achieve better peak performance. We also find the best performance among our experiments is achieved when pre-training the LLM with 11 billion tokens for 7 epochs. However, we can not find a good quality filter for the second strategy. We have tried to train a KenLM on WikiText as the quality filter for measuring the sample's quality in improving LLMs' general ability. But it does not work. From Figure 8, we find that strategies I and III can help the LLM to reduce the stability gap and achieve higher performance.

## F EFFECTIVENESS OF OUR STRATEGIES IN THE LEGAL DOMAIN

We consider strong baselines and report their legal performance in Table 5.

| Method | Training tokens number | MMLU-International-Law | MMLU-Professional-Law | Contract-QA | Avg |
|---|---|---|---|---|---|
| OpenLLaMa-3B | - | 27.1 | 28.4 | 51.0 | 35.5 |
| Full token baseline | 50B | 28.1 | 29.4 | 54.4 | 37.4 |
| Re-warming and re-decaying | 50B | 28.5 | 27.3 | 55.1 | 37.0 |
| Replay 10B data | 50B | 29.3 | 29.0 | 54.4 | 37.6 |
| Our strategies | 20B | **31.0** | **31.2** | **57.0** | **39.7** |

Table 5: Zero-shot accuracy across various legal benchmarks.

## G IMPACT OF LEARNING RATE AND TRAINING SUBSET SIZE

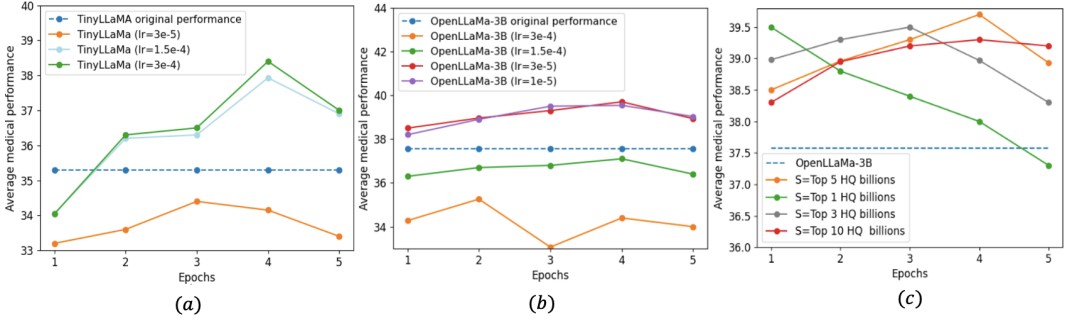

(a)                         (b)                         (c)

Figure 9: (a) reports the performance of TinyLlama-1.1B across multiple epochs. All these experiments use our strategies with different pre-training learning rates. (b) reports the performance of OpenLlama-3B across multiple epochs. All of the experiments in (a) and (b) use our strategies with different pre-training learning rates. (c) reports the performance of OpenLlama-3B across multiple epochs with different training subset sizes $S$. To collect the pre-training corpus with different sizes, we first rank all samples of the 50 billion medical tokens based on the perplexity calculated by the trained KenLM (see Sec. 3.1). Then, we select the first $S$ billion tokens with the lowest perplexity. For all experiments here, we report the average task performance of PubMedQA, MedMCQA, MMLU-medical-genetics, and MedQA-4-Option tasks.

**Impact of the learning rate** To analyze the influence of training factors like learning rate and training subset size, we conduct a series of experiments. We put the details in Appendix xxx. We find that too high learning rate leads to severe general-ability drops and too low leads to poor learning of new domain knowledge. Too large a subset (e.g., 10 billion tokens) results in a stability gap and slower performance, too small a subset yields better initial performance, but it also causes quick overfitting in later epochs. We further verify the best hyperparameter setup for our experiments. The pre-training learning rate is a crucial factor for updating LLMs during continual pre-training. To investigate its impact on our strategies, we conduct continual pre-training experiments with different learning rates. From Figure 9(a) and (b), we find that the optimal learning rate varies with the LLM scale: a small LLM (e.g., TinyLlama-1.1B) requires a higher learning rate (e.g., 3e-4), whereas larger LLMs (e.g., OpenLlama-3B) benefit from a lower learning rate (e.g., 3e-5). If the learning rate is too low (e.g., 3e-5 for TinyLlama-1.1B), the LLM cannot learn domain knowledge effectively

to boost performance. Conversely, if the learning rate is too high (e.g., 3e-4 for OpenLlama-3B), performance declines as the large learning rate leads to a significant plasticity gradient, causing the LLM to lose its general instruction-following ability for completing tasks. Based on our analysis experiments, we set the pre-training learning rate at 3e-4 for TinyLlama and 3e-5 for OpenLlama-3B's experiments.

**Impact of the training subset size** The size of the training subset is another important factor in our strategies. To determine the optimal training subset size, we conduct pre-training experiments on Llama-3b using various training subset sizes. From Figure 9(c), we observe that a smaller high-quality subset yields better initial performance and mitigates the stability gap (e.g., 1 billion tokens), but it also causes the performance to drop quickly in later epochs due to overfitting. A larger subset (e.g., 10 billion tokens) results in a stability gap and slower performance recovery, as the LLM needs to maintain a high plasticity gradient to learn a large number of new samples. Based on our experiments, we select a subset with 5 billion high-quality tokens, as it mitigates the stability gap, achieves the best peak performance, and is computationally effective.

# H    THE TRAINING DETAILS OF DEPLOYING OUR STRATEGIES INTO THE LLAMA-3 MODEL

**Pre-training details:** The pre-training task is to predict the next token with a context size of 8192. The training is executed using 16 H100 80GB GPUs. We employ the AdamW optimizer with $\beta_1 = 0.9, \beta_2 = 0.95$, a weight decay of 0.01, and a learning rate of 3e-5. We use a cosine learning rate scheduler with a 0.1 warmup ratio for gradual adaptation to training complexity and bf16 precision for computational efficiency. Gradient accumulation is set to 12 steps, and each training batch contains about 340 million tokens. We also add support for FlashAttention-2 (Dao, 2023) for more efficient inference and long-context decoding.

**Task-specific finetuning details:** We employ the AdamW optimizer with a weight decay of 0.01 and a learning rate of 3e-5. We use a cosine learning rate schedule with a 10% warmup ratio, decaying the final learning rate to 10% of the peak learning rate. We fine-tune the LLMs for 3 epochs. Since MMLU (Hendrycks et al., 2020a) does not have a training set, we follow (Chen et al., 2023b) and primarily consider the MMLU-Medical-Genetics benchmark, evaluating the model finetuned on MedMCQA.

For baselines in task-specific fine-tuning, we consider three kinds of baselines here: (1) Task-specific finetuning of the base model of open-source LLMs. This includes models such as Llama-2-70B, Llama-3-8B, and Llama3-Aloe-8B-Alpha (Gururajan et al., 2024). We copy their results from their respective papers (Gururajan et al., 2024) or the Meditron paper (Chen et al., 2023b) except for the Llama-3-8B, which we finetuned using the same process as our strategies. (2) Task-specific finetuning of continually pre-trained LLMs like meditron (Chen et al., 2023b), BioMistral SLERP 7B (Labrak et al., 2024), Llama-3-8B-full. These LLMs have been continually pre-trained with a medical corpus. We copy their results from their papers, except for Llama-3-8B-full, for which we continually pre-train the Llama-3-8B with 50B medical tokens collected in Section 3.1, and then finetune it using the same process as our strategies. (3) Closed-source LLMs. This includes models like ChatGPT and GPT-4 (OpenAI, 2023). The results are measured using the Microsoft Azure OpenAI API service (Shi et al., 2024).

**Instructions-tuning details:** We consider the combination of the question-answering training set of MedMCQA (Pal et al., 2022), MedQA (Jin et al., 2021a), PubMedQA (Jin et al., 2019), classification task HOC (Baker et al., 2016), relation extract task DDI2013 (Segura-Bedmar et al., 2013), inference task BioNLI (Bastan et al., 2022), and summarization task MIMIC-CXR (Johnson et al., 2019) tasks . To avoid potential data contamination, for each test sample of MedQA (Jin et al., 2021a), PubMedQA (Jin et al., 2019), and MedMCQA (Pal et al., 2022) tasks, we delete the training samples that contain its option. The specific dataset details are in Appendix C. For the training samples of theMedQA (Jin et al., 2021a),PubMedQA (Jin et al., 2019), and MedMCQA (Pal et al., 2022) tasks, we use the instruction template from the Meditron paper (Chen et al., 2023b). For the other datasets' training samples, we use their original instructions. We employ the AdamW optimizer with a weight decay of 0.01 and a learning rate of 3e-5. We use a cosine learning rate schedule with a 10% warmup ratio, decaying the final learning rate to 10% of the peak learning rate. We fine-tune the LLMs for 3

epochs. The global batch size is 96 and max sequence length is 1024. Unlike the above task-specific fine-tuning, we only tune one LLM here and use the instruction-tuned LLM to test all benchmarks.

For the baselines' results, we download the baselines' official models/deploy their APIs and then test their task performance using lm-eval-harnesses and Me-Llama's evaluation frameworks. If the paper does not release its model, we copy the results from the original paper (e.g., Me-Llama).

| Task type | Classification | Relation extraction | Natural Language Inference | Summarization |
|---|---|---|---|---|
| Datasets | HOC | DDI-2013 | BioNLI | MIMIC-CXR |
| Mistral-7B-instruct (Jiang et al., 2023) | 35.8 | 14.1 | 16.7 | 12.5 |
| Zephyr-7B-instruct-$\beta$ (Tunstall et al., 2023) | 26.1 | 19.4 | 19.9 | 10.5 |
| PMC-Llama-7B (Wu et al., 2023) | 18.4 | 14.7 | 15.9 | 13.9 |
| Medalpaca-13B (Han et al., 2023) | 24.6 | 5.8 | 16.4 | 1.0 |
| AlpaCare-13B (Zhang et al., 2023b) | 26.7 | 11.0 | 17.0 | 13.4 |
| BioMedGPT-LM 7B (Zhang et al., 2023a) | 23.4 | 15.5 | 17.9 | 6.2 |
| Me-Llama-13B (Xie et al., 2024b) | 33.5 | 21.4 | 19.5 | **40.0** |
| JSL-Med-Sft-Llama-3-8B (johnsnowlabs, 2024) | 25.6 | 19.7 | 16.6 | 13.8 |
| Llama-3-8B instruct | 31.0 | 15.1 | 18.8 | 10.3 |
| GPT-3.5-turbo-1106 | 54.5 | 21.6 | 31.7 | 13.5 |
| GPT-4 (OpenAI, 2023) | 60.2 | 29.2 | 57.8 | 15.2 |
| Llama-3-physician-8B instruct (ours) | **78.9** | **33.6** | **76.2** | 37.7 |

Table 6: Performance comparison for general medical tasks in the instruction-tuning setting. For BioNLI, DDI 2023, and HOC tasks, we report macro-F1. For MIMIC-CXR summarization tasks, we report Rouge-L as the result.

