# OpenReview forum: "Towards Efficient and No Forgetting Domain Continual Pretraining by Mitigating the Stability Gap"
_ICLR.cc/2025/Conference — ICLR 2025 Conference Withdrawn Submission_

### Official Review · Reviewer_5wq8 · 2024-10-28

**Soundness:** 2
**Presentation:** 1
**Contribution:** 2
**Rating:** 5
**Confidence:** 4

**Summary:**

This paper uses the concept of stability gap to explain the initial drop in LLM performance during continual pretraining in a new domain. The authors propose three training strategies to address the initial instability: 1) continually pretrain the LLM on a properly sized corpus subset for multiple epochs; 2) Continually pretrain LLM on a high-quality corpus subset; 3) Using data mixture rate that is similar to the pretraining data. The proposed strategies improve the accuracy of the LLM in the new domain when compared to the existing continual pretraining techniques.

**Strengths:**

- The proposed strategies are easy to implement.
- The LLM fine-tuned with the proposed strategies achieved the highest averaged accuracy score on a suite of medical question answering tasks.

**Weaknesses:**

## Major
- It is important to justify the methods of Muennighoff et al. (2024) and Lin et al. (2024) used in this paper. (I assume the four subsequent sentences explain the method (L158-161)). Here are some missing details:
  - Why was KenLM chosen?
  - What is a "high-quality medical reference corpus"? how do you define it? This is a fairly critical point because the "highest-quality" medical corpus can also be defined as those that resemble the downstream tasks the most, which makes the findings more expected (the closer the continual pretraining data is to the downstream tasks, the better the model will perform in the downstream tasks).
- The authors claim that the average accuracy of the LLM on the medical tasks initially drops and rises during the continual pretraining.
	- However, the drop itself does not look significant (less than 1% averaged accuracy). This makes the observation less strong. (See Question 2)
- This paper contains a flawed assumption due to the lack of access to the pretraining corpus. If a stability gap was proposed to explain the ability of the model to maintain performance on previous tasks, such an analysis cannot be achieved if we do not have access to the pretraining corpus.
	- The authors claimed (L233-235) that language modelling loss also preserves general knowledge and text modelling capabilities, which is a big assumption that is not backed by any evidence.
	- Note that text modelling capabilities may still be preserved via language modelling loss during the domain adaptation (continual) pretraining, however, we cannot guarantee that the general knowledge is still being preserved.
	- Additionally, there is no guarantee that the continual pretraining corpus was not included in the pretraining corpus. To examine this, the authors may have to conduct a pretraining from scratch.
- There exists a logical gap between the concepts of relative weight update, stability gradient, and instruction-following ability.
	- The authors concluded that the relative weight update indicates the stability gradient and, in turn, instruction-following ability (L241-253). However, there is no guarantee that relative weight update relates to stability gradient, let alone instruction-following capability.
        - Additional experiments using pretraining from scratch may help understand this phenomenon better.
- There are several mentions of a "properly sized" subset. However, they are not properly defined.
- The performance improvement (Figure 4) when compared to the baseline seems to be <1%. This does not look very significant.

## Minor
- Note that the submission and paper titles are different
- Abstract is generally filled with jargon which makes it harder to follow.
- L50-51: The last sentence of paragraph 1 in the Introduction can benefit from some citations.
- L56: Missing citation for "Previous research"
The introduction section still contains a lot of undefined jargon (i.e., "proper size", "highest-quality tokens", "data mixture")
- L194: Concluding that the "LLM has acquired medical domain knowledge" based on the perplexity score is a bit of an overclaim. Consider rephrasing it.
- Table 2: This misses the performance of the Llama-3-8B models without fine-tuning.
- The authors claim that the proposed strategies are computationally more efficient. By how much exactly? What metrics should you evaluate this on?

## Very minor (e.g., typos, etc)
- Use consistent verb tense (many inconsistent uses of present and past tenses)
- Typo L15: "phrase" -> "phase"
- L68: Instead of "harness" perhaps "mitigate" it? since you would like to mitigate the stability gap as opposed to harnessing it.
- Typo L125: lowercase "Language models"
- Typo L125: "RoBERTa"
- Page 4: Perhaps observations 1 and 2 can be swapped because in practice we may not know the downstream tasks during the (continual) pretraining phase.
- Figure 6b: The caption does not seem to be correct. The figure seems to show accuracy during law continual pretraining, while the caption is about relative parameter updates during the medical continual pretraining process.

**Questions:**

1) The stability gap concept proposed by previous studies is about the inability to maintain performance on **prior** tasks and the one mentioned in this paper is about the performance in the **new** target task. How are they two related in your experiments?
2) The initial drop in the averaged accuracy of the LLM on the medical tasks looks very insignificant.
	- Have you done a statistical test to verify this?
	- Is the small drop (<1%) in line with the findings of previous stability gap studies?
3) Data Mixture Results (Figure 4b and 4c):
	- The authors may need to compare the proposed strategies with the baseline (full data with multiple epochs).
	- The average medical and commonsense performance seems to drop in the 5th epoch. Why is that the case? What would happen if you continue the pretraining to 6th, 7th, ... epoch?
4) How similar is the "high-quality" medical reference corpus to the downstream tasks?
	- If you run the KenLM model on the downstream datasets, what is the perplexity? Would the perplexity be very low too?

---

### Official Review · Reviewer_X3mM · 2024-10-31

**Soundness:** 1
**Presentation:** 2
**Contribution:** 2
**Rating:** 3
**Confidence:** 4

**Summary:**

This paper explores the LLM behaviors when adapting to specific domains via continual pre-training. Authors point out unexpected "stability gap", which is an initial drop in performance before subsequent recovery. Authors provides three training strategies on this unexpected trend and conduct experiments on medical and law benchmarks.

**Strengths:**

1. The paper's motivation is clear, with well-structured on problems in continue pertaining, proposed strategies, and results.
2. Authors conducts experiments on different benchmarks across medical and law, to show the effectiveness of proposed methods in the continue-pretraining.

**Weaknesses:**

1. The proposed strategies seems similar with existing works conclusions. For examples, high quality data is important for model training [1,2] , using similar data mixture rate to the pre-training data to alleviate data distribution shift [3,4].
2. The experiments only conduct on relatively small models. The gap may be due to the the small model is not robust enough on the new dataset.  It is. unsure that if the larger models ( for example, 13B, 70B ) meet the same issue on continue pertaining.
3. The IFT model comparison is unfair to me due to some IFT models do not tuned on specific training dataset and they have different base models.
4. It is unsure that if the proposed IFT models is overfitting into the evaluation dataset by building IFT dataset based on original training data.


[1] Chen at al (2023). AlpaGasus: Training a Better Alpaca with Fewer Data
[2] Zhou et al (2023). LIMA: Less Is More for Alignment
[3]  Parmar et al (2024). Reuse, Don't Retrain: A Recipe for Continued Pretraining of Language Models.
[4] Ibrahim et al (2024). Simple and Scalable Strategies to Continually Pre-train Large Language Models

**Questions:**

1. I am not clear that how the high quality is obtained from original medical corpus.Can you further explain the quality evaluation metric for the data selection?
2. The figure 4 (a) is not clear to me. What's the x-axis represent? Can you further explain this Figure 4(a) and your finding?
3. The mixture strategy confused me.  Can you further explain the mixture strategies? Specifically,
"we follow the Llama mixture rate (Touvron et al., 2023a) to collect 5 billion tokens initially. We then replace the CC and C4 data (82% of the 5 billion tokens) with medical tokens sampled from the highest quality 5 billion medical tokens (HQ-5b). "
What's the initial 5 billion tokens ? How you further replace the token.
4. Is stability gap existing on larger models?  like 13B or larger models? Could you further conduct experiments on larger model to show the importance of the proposed issue?
5. Strategy 1 trains more epochs on smaller dataset may have higher chance to overfit. Can you further compare the continual training's performance on other OOD benchmark to show the overfitting issue (e.g. DROP, GSM8K, HumanEval etc).
6. Does the ' stability gap' changed by using different learning rate and warm-up strategies?

---

### Official Review · Reviewer_WWTv · 2024-11-02

**Soundness:** 3
**Presentation:** 3
**Contribution:** 2
**Rating:** 3
**Confidence:** 3

**Summary:**

The paper suggests that performance instability when training LLMs for specialized domains arises from distribution shifts. As such, they propose a new continual pre-training strategy that incorporates data quality and corpus distribution to identify "better" samples. In addition, the idea is to use these better subsets of samples and train for more epochs to ensure the LLM is in the performance recovery phase. The authors illustrate their performance on 4 benchmark QA datasets.

**Strengths:**

* The paper derives insights from stability gaps introduced in the context of visual models for continual learning to explain the behavior of performance drops with LLM continual pre-training for the specialized domain.
* Evaluation results with various biomedical-domain fine-tuned LLMs and QA datasets demonstrate the potential of the strategy.
* For some tasks and datasets, there is a noticeable improvement using less number of training tokens, especially on the MedMCQA task.

**Weaknesses:**

* The base architecture used is only the OpenLlama3B model with a single parameter size. The natural question is whether such a strategy is applicable across various LLM families and sizes (for example, GPT-NeoX was used by Ibrahim et al. with a 10B parameter model which might be comparable to the 8B rather than the 70B). Can you provide a comparison against GPT-NeoX 10B to provide a meaningful evaluation of your strategies?
* The motivation for the learning strategy is under a fixed computational budget, which seems to be only related to the number of training tokens and not the number of epochs. Can you explicitly define computational budget and then evaluate a scenario where token count and epoch count are kept constant to better understand the tradeoffs when considering a computational budget? This is a more elaborate setting than Section 3 which only assessed 5 epochs.
* The methodology, efficient learning strategies, and evaluation sections, all seemingly blend together without necessarily a coherent story or separation of sections. For example, in section 3, the differences between the two subsections seem to blend together whereas it would have been better to introduce the stability gap and demonstrate that the instability that is often observed seemingly is explained in the context of this, and should be done for one common set of experiments (note that there is swapping between medical domain, common sense task performance but with very little context for these experiments until Section 5). Section 4 seems to be more of an ablation study rolled in with their own method. As such, while it seems like the authors have done a lot of reasonable experiments, untangling what they are introducing and evaluating is very hard to understand without multiple reads. My suggestion would be to reorganize so that both section 3 and 4 are one contiguous section, where the first subsection focuses on motivating the stability gap in the context and then providing the strategy to mitigate this by choosing higher-quality samples. Section 5 can then focus on experiments where they are concisely targeting specific aspects of the strategy.
* There are a lot of results, but limited discussion about them, especially comparison of performance. Please provide a more detailed discussion of your results. Moreover, it would be helpful to clarify which fine-tuned models may not be tuned on the same task so the performance might be hindered by this, whereas others might be fine-tuned on the task so it might be reasonable to expect them to do well. To accommodate this expansion, space can be made by shrinking some of the figures.
 * Some of the graphs do not provide sufficiently more information. For example Figure 2 (b) reports the beginning of only 1 model for the millions of tokens, but the trend doesn't seem to be that much more informative than Figure 2a. Similarly, much of the motivation was for specialized domain but there is only a focus on medical domain whereas it would have been more compelling with Appendix B results embedded here.

**Questions:**

* Why is the performance substantially better for your strategy on MedMCQA? The tasks, performance gains seem more mixed and not necessarily as beneficial. What about MedMCQA benchmark makes it benefit the most from the continual pre-training?
* Does this technique work for other datasets? In looking at the legal dataset results in Appendix F, there are similar findings suggested for the zero-shot but the experimental comparisons.

---

### Official Review · Reviewer_KEZA · 2024-11-04

**Soundness:** 3
**Presentation:** 3
**Contribution:** 2
**Rating:** 5
**Confidence:** 5

**Summary:**

The manuscript focuses on the problem of stability gap- i.e., LLMs dropping their performance drastically when continually pretrained on a new domain and then recovering performance gradually. The manuscript demonstrates stability gap using the medical domain using relatively smaller language model and proposes (three) strategies to overcome and stabilize pre-training loss- (1) continual pre-training with a random partition of the domain across multiple epochs (2) continual pre-training using a notion of high-quality tokens selected using KenLM (3) Utilizing existing pre-training data-mixture ratios to selectively replace the current corpora with target domain corpora. The manuscript then applies the strategy to Llama-3-8B- in continually pretrain and fine-tuning settings.

**Strengths:**

- The paper is reasonably well organized and written.
- The findings are well explained and justified with empirical analyses where required.
- The authors conduct extensive experimentation to cover different possible research questions

- The concept of stability gap is not new and has been extensively studied in Computer vision but relatively less in NLP. The paper draws it's research question from this and possible solutions from CV. The paper compares its proposed strategies with existing work, e.g. Ibrahim et al (Rewarm and decay), Replay (Chen et al) etc.

**Weaknesses:**

- The paper uses DEITA and KenLM for assessing the quality of samples in the target domain.

 Need a baseline with only Continually pretrained with all data (all data vs only 50B) vs proposed strategy
- Table -1: The	performance vs 10B replay is pretty close. The performance difference seems to solely arise due to MedMCQA;
 may need statistical significance tests to see if the differences are due to proposed strategies or due to randomness.
- Table 2: >20% performance jump again on MedMCQA for Physician vs LLaMa-3-8B Fine-tuned seems odd. Are there any possible explanations, especially the difference in performance for other datasets <5%. (Please add statistical significance tests- see last bullet)
-	Performance could be possibly validated using statistical significance tests- either using permutation or signed rank tests. see- https://aclanthology.org/D12-1091/

**Questions:**

- Line 288: `... of each sample in the entire medical corpus.' what does each sample indicate (documents, QA pairs or anything else?)? Are the samples drawn from the only the dataset being evaluated or all of them combined?

- How was the 50B domain text obtained from wiki-medical-terms? The website seems to indicate that the corpus has 6k medical terms and their descriptions. Does the whole terms + description have 50B tokens? Any other relevant statistics?
- The paper's main contribution seems to arise due to creating the High Quality (HQ) partition using KenLM- Could the authors add more information about how this was performed? For e.g., what were size of _n_ if an n-gram-based approach was used?
- Creating HQ partition could have been done in other ways- entropy, ranking or using MLE for importance. Can the authors comment why KenLM was chosen. Can they compare this selection with others? Do they work in similar ways/show similar performance?

---

### Note · Authors · 2024-12-16

I have read and agree with the venue's withdrawal policy on behalf of myself and my co-authors.